# Quantitative Evaluation of In Vivo Corneal Biomechanical Properties after SMILE and FLEx Surgery by Acoustic Radiation Force Optical Coherence Elastography

**DOI:** 10.3390/s23010181

**Published:** 2022-12-24

**Authors:** Yanzhi Zhao, Yirui Zhu, Yongbo Wang, Hongwei Yang, Xingdao He, Tomas Gomez Alvarez-Arenas, Yingjie Li, Guofu Huang

**Affiliations:** 1School of Medical, Nanchang University, Nanchang 330031, China; 2Department of Ophthalmology, The Third Affiliated Hospital of Nanchang University, Nanchang 330008, China; 3School of Physics, Nanjing University, Nanjing 210093, China; 4Key Laboratory of Opto-Electronic Information Science and Technology of Jiangxi Province, Jiangxi Engineering Laboratory for Optoelectronics Testing Technology, Nanchang Hangkong University, Nanchang 330063, China; 5Institute for Physical and Information Technologies, Spanish National Research Council, Serrano 144, 28006 Madrid, Spain

**Keywords:** SMILE surgery, FLEx surgery, optical coherence elastography, optical coherence tomography, acoustic radiation force, corneal biomechanical properties

## Abstract

The purpose of this study is to quantitatively evaluate the differences in corneal biomechanics after SMILE and FLEx surgery using an acoustic radiation force optical coherence elastography system (ARF-OCE) and to analyze the effect of the corneal cap on the integrity of corneal biomechanical properties. A custom ring array ultrasound transducer is used to excite corneal tissue to produce Lamb waves. Depth-resolved elastic modulus images of the in vivo cornea after refractive surgery were obtained based on the phase velocity of the Lamb wave. After refractive surgery, the average elastic modulus of the corneal flap decreased (71.7 ± 24.6 kPa), while the elastic modulus of the corneal cap increased (219.5 ± 54.9 kPa). The average elastic modulus of residual stromal bed (RSB) was increased after surgery, and the value after FLEx (305.8 ± 48.5 kPa) was significantly higher than that of SMILE (221.3 ± 43.2 kPa). Compared with FLEx, SMILE preserved most of the anterior stroma with less change in corneal biomechanics, which indicated that SMILE has an advantage in preserving the integrity of the corneal biomechanical properties. Therefore, the biomechanical properties of the cornea obtained by the ARF-OCE system may be one of the essential indicators for evaluating the safety of refractive surgery.

## 1. Introduction

The introduction of the femtosecond laser has facilitated the development of corneal refractive surgery [1,2]. The femtosecond lenticule extraction (FLEx) procedure was first performed by Sekundo in 2008 using the Visumax femtosecond system (Sekundo, et al., 2008). Both the flap and the refractive lenticule were created in a single surgery using the femtosecond laser, which shortens the procedure time compared with previous refractive surgery methods. In 2011, Sekundo optimized the FLEx procedure by reducing the corneal incision to 2–3 mm and preserving the more mechanically robust anterior stroma. The optimized FLEx procedure was then named small incision lenticule extraction (SMILE) [3]. SMILE was approved by the FDA in 2016 and has since been widely adopted by refractive surgeons around the world.

Compared with flap-based refractive surgery, the cap and residual stromal bed (RSB) of the cornea after SMILE surgery share stress distribution. Theoretically, SMILE has the advantage of preserving corneal biomechanics, which has been validated in numerous clinical and experimental studies [4,5,6,7]. However, most studies examined corneal biomechanics using the Corvis ST and the Ocular Response Analyzer (ORA); the results of both instruments showed that there was no difference between SMILE and FLEx [8]. The possible reason is that the two instruments are highly influenced by corneal thickness and intraocular pressure and cannot detect slight differences caused by structural changes in the cornea [9]. Therefore, more advanced testing equipment is needed to study the effects of different surgical approaches on corneal biomechanics.

For the in vivo measurement of the corneal biomechanical properties, elastography techniques were developed, such as Brillouin microscopy and ultrasound elastography. Brillouin microscopy was used to measure the compression modulus of the cornea with a high resolution based on the longitudinal wave speed. However, Brillouin signal acquisition requires long scanning times, which limits the clinical translation of Brillouin microscopy for the detection of corneal elasticity in the human eye [10]. Ultrasound elastography has been used to quantify the elastic modulus of ocular tissues, such as the cornea and retina. However, the resolution of ultrasound elastography is on the sub-millimeter scale, and there are limitations in quantifying the elasticity of the stratified structure of the human cornea [11,12,13]. Similar to ultrasound elastography, acoustic radiation force optical coherence elastography (ARF-OCE) is a new extension technology based on optical coherence tomography (OCT) that can quantitatively measure the elastic modulus of biological tissue with the advantages of high-resolution, high-sensitivity, and rapid three-dimensional image. The elastic modulus of biological tissue can present the corresponding stress distribution information, which has the advantages of high detection speed, high 3D spatial resolution, and high vibration sensitivity. OCE has led to a wide range of applications in ophthalmology since its first demonstration in biology twenty years ago [14,15]. Han et al. proposed a modified Rayleigh-Lamb mode OCE system to analyze corneal viscoelasticity [16]. Qu et al. assessed the biomechanical properties of the rabbit cornea using acoustic radiation force OCE [17]. Larin et al. systematically studied the spatial characterization of corneal biomechanical properties with parameters that UV cross-linking, corneal hydration, intraocular pressure (IOP), and corneal thickness combine the air puff OCE system and phase velocity dispersion relationship [18,19,20,21]. There are also some studies on the measurement of the elasticity of the human cornea using OCE. For instance, compression swept-source OCE was used to assess corneal displacement at different depths [22]. Ramier et al. devised a contact transducer probe-based OCE system that was able to measure the shear modulus of the human cornea in vivo [23]. The effects of heartbeat and respiration on the measurement of corneal biomechanics were analyzed by an air-puff-based OCE system [24]. However, to the best of our knowledge, few biomechanical studies have been carried out after corneal refractive surgery with the OCE system.

This is the first study to assess the differences in corneal biomechanics after SMILE and FLEx using a customized acoustic radiation force-based OCE system and phase velocity of the Lamb wave model. The elastic modulus of the cornea after refractive surgery was measured using in vivo rabbit models.

## 2. Materials and Methods

### 2.1. Subjects and Surgical Techniques

This study was approved by The Ethics Committee of Sun Yat-sen University Zhongshan Eye Center Nanchang Eye Hospital. Twelve healthy New Zealand white rabbits (age: 4–5 months, weight: 3.5–4 kg) were randomly divided into the SMILE group (*n* = 6) and the FLEx group (*n* = 6). One eye of each rabbit was randomly selected to perform the refractive surgery. Before the refractive surgery and OCE measurement, the anesthesia procedure was performed by intramuscular injection of Ketamine. In order to prevent the rabbit’s third eyelid from interfering with the operation, the third eyelid was dissected before the operation. All refractive surgery procedures were completed by an experienced surgeon (GH) using the Visumax (500 kHz; Carl Zeiss Meditec, Jena, Germany) femtosecond laser platform. The following surgical parameters were used in both the FLEx group and the SMILE group: flap/cap diameter 7 mm; flap/cap thickness 110 µm; lenticule diameter 6 mm; lenticule thickness 6D (approximately 100 μm); spot energy < 200 nj. The only difference between the two groups was the length of the corneal incision for the removal of the lenticule. Specifically, a 30-degree corneal incision was used in the SMILE group, while a 300-degree incision was made in the FLEx group. After the procedure, the eyes were treated with antibiotic levofloxacin eye drops (Santen, Japan) 4 times a day for 7 days and 0.1% fluorometholone eye drops (Santen, Japan) 4 times a day for 30 days.

### 2.2. Pre-Operative and Post-Operative Examinations

Prior to the refractive surgery, ophthalmological examinations were performed on the rabbits using ultrasound pachymetry, a slit-lamp, Covis-ST, and Pentacam corneal topography to determine whether they met surgical requirements. According to the incisions of SMILE and FLEx, an experiment protocol was designed to accurately assess the differences in corneal biomechanical properties, as shown in Figure 1. The red dotted line shows the imaging area, and the black solid arc is the incision made in the SMILE surgery and the corneal flap in the FLEx surgery.

### 2.3. Design of the Acoustic Radiation Force Optical Coherence Elastography System

For the in vivo measurement of corneal biomechanical properties before and after refractive surgery, an acoustic radiation force-based OCE system was devised, as shown in Figure 2. It allows us to detect the propagation of elastic waves using an acoustic radiation force swept-source OCE system with a 1310 nm central wavelength, 50 kHz A line rate, and 10 μm axial and 15 μm lateral spatial resolution in air. The parameter of the phase stability is 175 mrad, which corresponds to a displacement sensitivity of 13 nm. The ARF-OCE system through the scan lens focuses on the rabbit cornea with 10 mW of output power, which is below the threshold value specified in ANSI Z136.1. A customized 4.5 MHz central frequency focused ring array ultrasound transducer with eight array elements was designed to excite the cornea tissue with a 4 cm focal length and a focal range of 1 mm by 1 mm, which allows for localized ARF-induced displacement or Lamb waves in the lateral direction, as shown in Figure 2. The function generator and bandwidth amplifier, as the controllers of the ultrasound transducer, were synchronized with the image acquisition system. M-B mode elastography and B mode structure imaging of the corneal tissue were performed using a two-dimensional (2D) galvanometer that was driven by a C++ system. Briefly, the direction of acoustic radiation force was perpendicular to the direction of OCT detection light (i.e., the depth direction *z*). Consecutive A-scans at a single point on the *x*-axis (fast scan) yield a total of 500 M-scans, then the 2D galvo-mirror moved the OCT beam (slow scan) to the next point on the *x*-axis to perform the same M-scan. A total of 1000 points at different positions on the *x*-axis constituted the B scan with a total process duration of 10 s. The accuracy of our developed ARF-OCE system for elastography has been evaluated in previous work [25,26]. During the OCE experiment, the rabbit eyeball was squeezed within a rubber drape that serves as a container to immerse the eyeball in phosphate-buffered saline (PBS).

### 2.4. Quantification of the Elastic Modulus

Here, we measured the corneal elastic modulus using a frequency-dependent phase velocity of the Lamb wave. The phase change *p* of the optical signal is a function of the Lamb wave vibration displacement *D* [27,28,29], according to the phase-resolved Doppler variance arithmetic:(1)$$D=\frac{\lambda p}{4\pi n}$$
where, *λ* is the central wavelength of the swept source laser in the ARF-OCE system, and *n* is the refractive index of the air. The vibration moves to different positions with the propagation of the Lamb wave, the phase delay of the OCE system can be quantified using Equation (2):(2)$$\;\frac{\lambda_{L}}{L}=\frac{2\pi }{\Delta \phi }$$
where, $\lambda_{L}$ is the wavelength of the Lamb wave, *L* is the propagation distance, and  Δϕ is the phase delay. The phase velocity *C* of the Lamb wave can be calculated by the phase delay Δϕ using Equation (3):(3)$$C=2\pi f\;\times \;\frac{L}{\Delta \phi \;}$$
where, *f* is the frequency of Lamb wave, which can be obtained by 2D Fourier transform of the spatial-temporal displacement. Furthermore, the biomechanical properties of the cornea can be quantified according to the relationship between the velocity and the Elastic modulus (Young’s modulus):(4)$$E=\frac{9\rho \times C^{4}}{{\left(\pi \times f\times h\right)}^{2}}$$

Here, ρ is the density of the cornea, which is 1062 kg/m^3^, and *h* is the corneal thickness.

### 2.5. Statistical Analysis

The ARF-OCE experiments were performed twice: 10 days before the surgery and 30 days after the surgery. Statistical analysis was carried out using R software (version 3.5.3, Lucent Technologies). A student’s *t*-test was used to analyze the difference in biomechanical properties of the corneal cap, flap, and RSB between FLEx and SMILE. *p* < 0.05 was considered as statistical significance.

## 3. Results

### 3.1. Central Corneal Thickness

Central corneal thickness (CCT) was measured by ultrasound pachymetry for the elastic modulus calculation of the cornea and is shown in Table 1. Before the refractive surgery, the average CCT of the SMILE and FLEx groups were 344 ± 4.6 µm and 347 ± 10 µm, respectively. After the refractive surgery, the mean CCT of the SMILE and FLEx groups were 238 ± 7.1 µm and 238 ± 9.4 µm. Statistical results showed that there is a statistically significant difference in CCT values after SMILE (*p* < 0.05) and FLEx surgery (*p* < 0.05).

### 3.2. ARF-OCE Results of Normal Cornea

To determine the changes in the biomechanical properties of the cornea before and after refractive surgery, the OCE experiment was performed before refractive surgery to measure the biomechanical properties of the intact cornea. Figure 3a showed the images of the different corneal layers along the depth direction that were obtained, including the epithelium, Bowman’s layer, stroma Descemet’s membrane, and endothelium. When the acoustic radiation force excited the corneal tissue, the vibration signal was detected by the OCE system and mapped with different colors. As shown in Figure 3b, the blue and red are vibration signals in the upward and downward directions, respectively. Furthermore, the spatial-temporal propagation of the Lamb wave at a selected depth was obtained by re-slicing the Doppler B-scan image, as indicated by the yellow dashed line shown in Figure 3c. The group velocity of the Lamb wave was calculated based on the distance and the detection time.

Considering the anisotropic material properties of the cornea, the Lamb wave phase velocity was further used to analyze the biomechanical properties of the cornea. Since the phase velocity depends on the frequency-dispersion relationship, the vibrational curves in the frequency-wavenumber domain obtained by the 2D Fourier transform of the spatial-temporal displacement are shown in Figure 4a, and the phase velocity of the Lamb wave was calculated using C=2πf / k (Figure 4b). Based on the mapping of the phase velocity, the elastography of the cornea at all available depths can be obtained by repeating the above steps. For each layer of the intact cornea, a central corneal region was selected for depth-resolved elastography (Figure 4c) by repeating the above program. The elastic modulus of the cornea decreased from the anterior stroma to the endothelium, with a range of 149–86 kPa, which was consistent with the results of OCE and ultrasonic elastography [30,31]. The maximum elastic modulus of the intact cornea was 144 ± 5 kPa in the anterior stroma and the minimum value was 89 ± 3 kPa in the endothelium.

### 3.3. ARF-OCE Results of the FLEx and SMILE Surgery

In this study, we quantitatively compared the biomechanical properties of the cornea after flap-based and cap-based surgeries. For the FLEx and SMILE surgery groups, one postoperative cornea was selected from each group to describe the ARF-OCE results in detail. After the FLEx procedure, the surface morphology of the cornea obtained by the slit-lamp is shown in Figure 5a, and the red star indicates the flap location. The corneal flap and RSB after the FLEx procedure was imaged, Figure 5b shows the 3D reconstructed structure, and Figure 5c shows the 2D tomographic structure of the red dotted line in Figure 5b. The red arrow indicates the boundary of the flap and the RSB in Figure 5c.

For the quantitative assessment of the biomechanical properties of the cornea after FLEx, spatial-temporal displacement of the Lamb wave was first obtained (Figure 5d). According to the same calculation method described above, the elastography along the depth direction was obtained, and the result is shown in Figure 5e. The average elastic modulus of the RSB was 290 ± 5 kPa, which is the layer with the largest value after FLEx. The average elastic modulus of the corneal flap was 63 ± 2 kPa, which is the layer with the smallest value.

In the SMILE group, after the 6-mm diameter stromal lenticule was removed through a small incision, the Bowman’s layer was preserved. Figure 6a is the surface morphology as shown by the slit-lamp. The red star indicates the incision location. The red dotted ellipse represents the location of the incision in Figure 6b, and the red dotted line indicates the location of the 2D structure; the result is shown in Figure 6c. The red arrow indicates the boundary of the cap and RSB in Figure 6c. Figure 6d shows the spatial-temporal displacement of the Lamb wave in the SMILE cornea. Furthermore, the results of depth-resolved corneal elastography are shown in Figure 6e. The average elastic modulus of the corneal cap and RSB were 213 ± 3 kPa and 224 ± 4 kPa, respectively.

Figure 7 shows the statistical results of elastic modulus of the normal cornea, corneal cap/flap, and RSB before and after the FLEx and SMILE surgery. For the normal cornea, the average elastic modulus was 91.7 ± 28.1 kPa. For the FLEx group, the average elastic modulus of the corneal flap and RSB were 71.7 (±24.6) kPa and 305.8 (±48.5) kPa, respectively. For the SMILE group, the average elastic modulus of the corneal cap and RSB were 219.5 (±54.9) kPa and 221.5 (±43.2) kPa, respectively. The differences between the SMILE and FLEx groups were statistically significant (*p* = 1.7 × 10^−7^).

## 4. Discussion

In this study, the corneal elastic modulus along the depth direction was quantified after SMILE and FLEx using the ARF-OCE system in order to evaluate the biomechanical properties of the corneal cap, flap, and RSB. The results showed that compared with FLEx, SMILE preserved most of the anterior stroma with less change in the biomechanical properties of the cornea. Therefore, SMILE may have an advantage in preserving the integrity of the corneal biomechanical properties.

Although computer models and finite element simulations have confirmed that the SMILE procedure has the potential to preserve corneal biomechanics over flap-based refractive procedures, there is still some controversy due to the large differences in clinical studies [7,8,32,33]. For the in vivo measurement of corneal biomechanics, common commercial devices currently used clinically are ORA and Covis-ST. These are two different pneumatic applanation devices that provide an overall qualitative analysis of the biomechanical properties of the cornea; however, they do not provide depth-resolved characterizations regarding different layers of the cornea. Thus, changes in corneal biomechanical properties due to structural changes in the flap, cap, and RSB after refractive surgery cannot be precisely quantified. The detection and calculation methods used in this study, which combined the nondestructive ARF-OCE system with the phase velocity of the Lamb wave model (Figure 3a,b), have been proven to be effective for elastography of eye tissues and have great potential for clinical application [18,19,20,21]. The amplitude of the Lamb wave was close to 30 µm, and the axial resolution of the ARF-OCE system was 10 µm, which enables high-resolution imaging of the Lamb wave propagation process. The phase velocities of the Lamb wave at different depths can be quantified separately in the frequency domain. Furthermore, the corneal elastic modulus can be calculated hierarchically based on the relationship between the Lamb wave phase velocity and the corneal elastic modulus, that is, the elastic modulus can be quantified separately for the postoperative corneal flap/cap and RSB. The proposed method addresses the clinical limitations in the quantitative assessment of corneal elastic modulus in regional stratification.

The anterior third stroma of the cornea has higher mechanical strength than the posterior [34]. The experimental results of the ARF-OCE system showed that the elastic modulus of intact cornea decreased gradually from the anterior stroma to the endothelium (Figure 4c). SMILE maintains a stronger anterior stromal compared with FLEx. Thus, SMILE theoretically facilitates the stabilization of corneal biomechanics after the surgery. Damgaard et al. compared the differences in corneal biomechanics between 110 µm and 160 µm cap thickness after SMILE. The results showed that a correction in the deeper stromal is beneficial for maintaining corneal biomechanical strength [35], and the finding is consistent with that of a mathematical biomechanical model [32]. However, in Laser in situ Keratomileusis (LASIK) surgery, thicker flaps reduce postoperative corneal biomechanical properties. Sinha et al. created a finite element model of the cornea and modeled the material properties as a function of depth to analyze corneal biomechanical properties after LASIK and SMILE [36]. The results showed that the corneal stress distribution after SMILE was similar to the geometric-matched model, while the corneal flap stress decreased and the stress of the stromal bed increased after LASIK. The results of the Brillouin microscopy-based approach to study the biomechanical properties of the cornea after LASIK surgery showed that LASIK flap creation significantly reduced Brillouin shift in the anterior third of the stroma in porcine eyes [37]. Compared with the normal cornea, the post-LASIK cornea has a lower Brillouin shift (lower corneal stiffness) across the total corneal thickness, as well as within the anterior and central [38,39,40]. Corneal biomechanics play a key role in corneal geometry [41,42]. Higher mechanical strength allows the cornea to carry high stress per unit volume. At the same IOP, corneas with more severely damaged biomechanical properties underwent greater deformation after refractive surgery, and the biomechanical properties were below the threshold required to maintain the shape, which can lead to iatrogenic keratectasia. In our study, the elastic modulus of the corneal cap increased after SMILE, which was different from the decrease in the elastic modulus of the flap after FLEx (Figure 7). In addition, the distribution of corneal elastic modulus after SMILE (Figure 6e) was in a similar trend to that of the intact cornea (Figure 4c), and it is beneficial to reduce the risk of corneal ectasia after SMILE.

Furthermore, the structural integrity of the cornea is directly related to its mechanical strength, and both the corneal flap and cap in refractive surgery reduce the biomechanical properties. In this study, the flap created by FLEx required cutting most of the anterior stroma, while SMILE removed the lens through a small 2 mm incision, preserving most of the anterior stroma (Figure 6a,b). Bryant et al. reported a 49% reduction in the stiffness of stromal collagen fibers in the flap area after FLEx compared with the corneal cap after SMILE [33]. 2D-extensometry of the human and porcine cornea showed that, compared with FS-LASIK and FLEx, the SMILE procedure resulted in a superior biomechanical stability of the cornea [4,43]. In our ARF-OCE study, the corneal elastic modulus increases with the increase in unit stress, and the results showed that the elastic modulus of the cornea increased after SMILE, the average elastic modulus of the corneal cap and RSB were 219.5 (±54.9) kPa and 221.5 (±43.2) kPa, respectively. The elastic modulus of the corneal flap after FLEx was significantly lower than that of the corneal cap after SMILE (*p* = 0.0001), and the elastic modulus of the corneal RSB after the FLEx procedure was significantly larger than that after SMILE (*p* = 0.017). The corneal flap after FLEx cannot carry the same pressure as before the surgery. The stress was redistributed on the cornea and reached a steady state, which was borne by the RSB. After SMILE, the anterior stroma is only slightly damaged, and the corneal cap and RSB share the redistributed pressure.

One limitation of this study is that it did not consider the effect of physiologic variations in IOP on the elastic modulus measurements. However, the range (10–13 mmHg) of variation in IOP under physiologic conditions is small, so it does not affect the conclusions of this study [44]. Another limitation of this study is that corneal geometry and anisotropy can affect the center frequency of the Lamb wave, leading to certain errors in the corneal elastic modulus. In existing studies, the reported corneal shear modulus based on the shear wave OCE system was 1.8–52.3 kPa, which is consistent with our experiment results (Figure 6c) [17,20]. However, the study of tension or inflation experiments found that the corneal elastic modulus was much larger than the results of OCE experiments [45,46]. Obviously, corneal shear, tensile, and inflation strains exhibit different corneal behaviors, and, thus, corneal biomechanics are anisotropic. Considering the microstructural characteristics of the cornea, a more accurate biomechanical model needs to be developed to assess the anisotropic elastic modulus [47]. This is beyond the scope of this study but will be considered in future studies. In addition, we performed the ARF-OCE experiment in the first month after surgery. Postoperative corneal biomechanical properties should be continuously monitored, and corneal biomechanical changes should be monitored over time.

In summary, the ARF-OCE system was first used to measure the biomechanical properties of the cornea after SMILE and FLEx. The elastic modulus of the corneal cap/flap and RSB were calculated. The experiment results showed that both the SMILE and FLEx procedures will alter the biomechanical properties of the corneal, leading to a redistribution of corneal stress. However, both the corneal cap and the RSB carry the IOP, and, thus, SMILE has an advantage in preserving the integrity and stability of the biomechanical properties of the cornea. Quantifying the biomechanical properties of the cornea after refractive surgery allows us to identify the optimal surgical method to preserve the biomechanical properties of the cornea and reduce the risk of iatrogenic keratectasia [48].

## Figures and Tables

**Figure 1 sensors-23-00181-f001:**
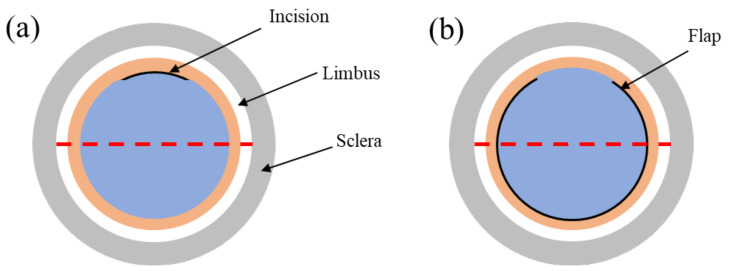
The ARF-OCE experiment protocol of the SMILE and FLEx surgery, (**a**) is the SMILE surgery incision structure, the SMILE procedure preserves most of the anterior corneal stroma by creating a small 2–3 mm incision in the anterior corneal stroma and removing the stromal microcrystalline lens; (**b**) is the FLEx surgery flap structure, the FLEx procedure uses a femtosecond laser to create a corneal flap on the surface of the cornea and then thin the stroma.

**Figure 2 sensors-23-00181-f002:**
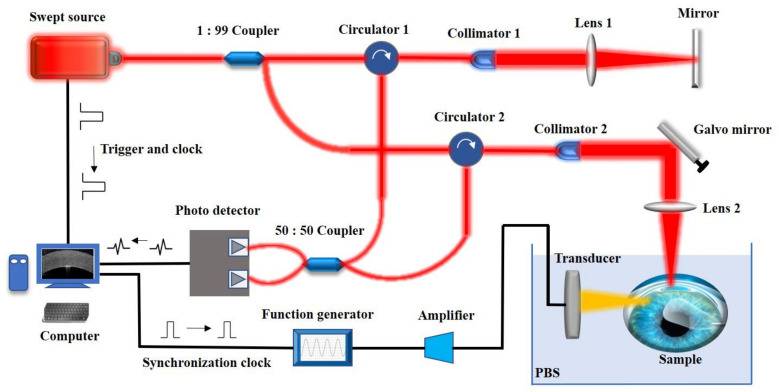
The acoustic radiation force based OCE system diagram. The swept laser with a central wavelength of 1310 nm is split by a 1:99 optical coupler and then enters the reference arm and the sample arm, respectively, and the interference signal is detected by an optical balance detector. The laser provides the sampling trigger signal and sampling clock signal. The synchronization control signal of the acoustic radiation force excitation is generated by the computer and synchronized with the OCT sampling clock.

**Figure 3 sensors-23-00181-f003:**
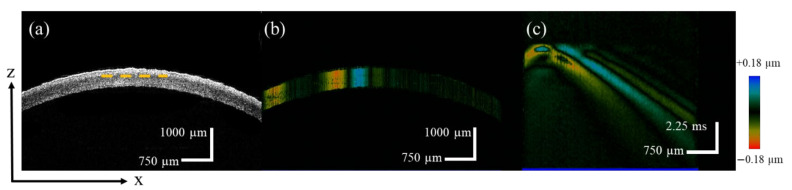
The ARF-OCE experiment results of the normal cornea, (**a**) is the structure image of the cornea in OCT B-scan, (**b**) is the vibration mapping in the OCE B-scan, and (**c**) is the spatial-temporal displacement diagram of the Lamb wave.

**Figure 4 sensors-23-00181-f004:**
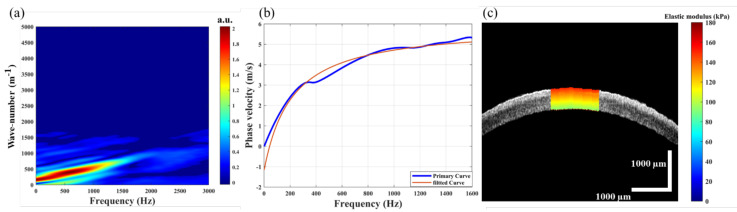
The phase velocity calculation of the Lamb wave in normal cornea, (**a**) is the wave number-frequency map of the Lamb wave, (**b**) is the phase velocity curve depends on the frequency, and (**c**) is the depth-resolved elastography result of the normal cornea.

**Figure 5 sensors-23-00181-f005:**
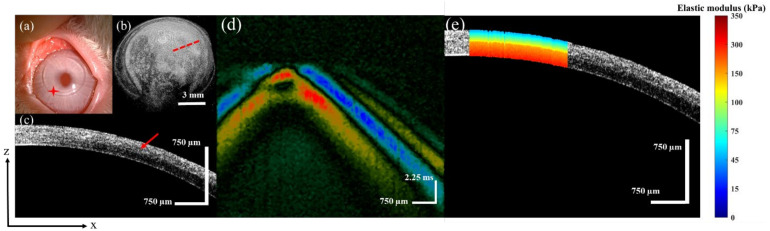
The FLEx surgery corneal images, (**a**) is the FLEx surgery surface morphology, (**b**) is the 3D reconstruction structure, (**c**) is the 2D tomography, (**d**) is the spatial-temporal displacement diagram of the Lamb wave, (**e**) is depth-resolved elastography result of the cornea.

**Figure 6 sensors-23-00181-f006:**
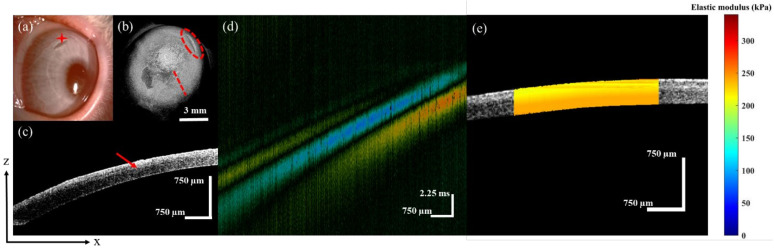
The SMILE surgery corneal images, (**a**) is the SMILE surgery surface morphology, (**b**) is the 3D reconstruction structure, (**c**) is the 2D tomography, (**d**) is the spatial-temporal displacement diagram of the Lamb wave, and (**e**) is depth-resolved elastography result of the cornea.

**Figure 7 sensors-23-00181-f007:**
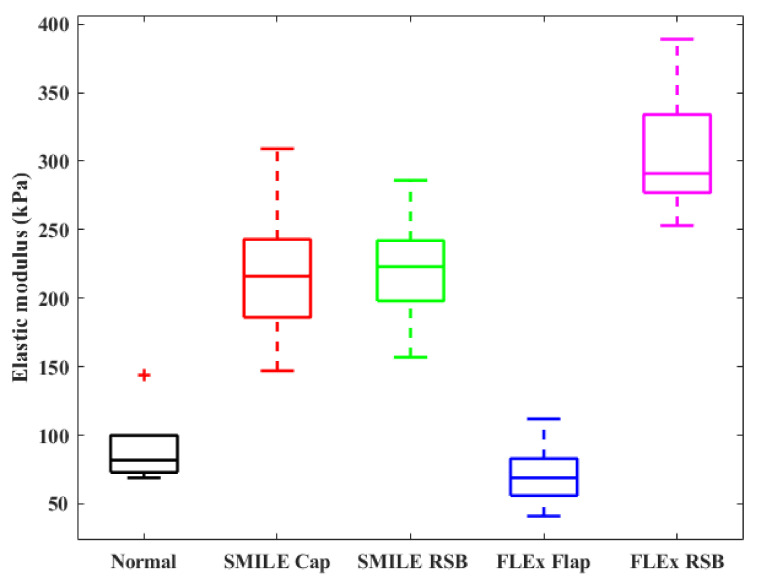
Elastic modulus statistical results of the normal cornea, corneal cap/fap and RSB before and after FLEx and SMILE surgery.

**Table 1 sensors-23-00181-t001:** Central corneal thickness before and after SMILE and FLEx surgery respectively.

Data (Mean ± SD)	SMILE (*n* = 6)	FLEX (*n* = 6)
CCT (Pre-), µm	344 ± 4.6	347 ± 10
CCT (Post-), µm	238 ± 7.1	238 ± 9.4
*p*-value	4.4985 × 10^−9^	1.3300 × 10^−9^

CCT = central corneal thickness; SMILE = small incision lenticule extraction; FLEx = femtosecond lenticule extraction; SD = standard deviation; *p* < 0.05 is considered statistically significant.

## Data Availability

The raw data presented in the study further inquiries can be directed to the corresponding author.

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
