# Peer review of "Quantitative Evaluation of In Vivo Corneal Biomechanical Properties after SMILE and FLEx Surgery by Acoustic Radiation Force Optical Coherence Elastography"

_sensors, 2022, doi:10.3390/s23010181_

Round 1

Reviewer 1 Report

Zhao and colleagues evaluated corneal biomechanical properties following femtosecond laser-assisted corneal refractive surgery in a rabbit model using acoustic radiation force optical coherence elastography. The methods are appropriate and clear and the results support the conclusions. The manuscript has been formatted and written very well. Although the outcome was expected theoretically and based on mathematical models, this study provides additional data using the ARF-OCE system that adds to the current knowledge.

General comment:

The relationship between the elastic modulus (E) and the tissue stiffness requires further explanation and clarification. For instance, Fig. 4 shows a decreasing E towards the posterior stroma. Since we know that the anterior stroma has higher mechanical strength, how would you explain this finding?

Major comments:

1.     Please briefly discuss the advantages and disadvantages of the ARF-OCE compared with Brillouin microscopy and other in vivo elastography techniques in the Introduction (e.g., methods, reliability, feasibility, potential application in the human eye, …).

2.     Is there any test-retest variability data for the device used in this study?

3.     I suggest using a similar color coding for Figures 4–6 so that the changes can be compared between the three models.

Minor comments:

1.     Abstract, line 3: “… completeness of corneal biomechanical properties”; completeness should be replaced with a more appropriate word.

2.     Abstract, line 8: RSB should be defined on the first appearance in the abstract.

3.     Introduction, paragraph 1, line 5: “Shortened operation time compared to previous methods of refractive surgery”, this sentence is incomplete. Please revise.

4.     Section 2.2., line 12: please add the approximate central lenticule thickness as well.

5.     Please add the relevant explanations to the legends of Figures 1 and 2.  

6.     Figure 2. Please define PBS.

7.     Section 3.3., line 5: “SLP” is not a known abbreviation for the slit-lamp and can be removed.

8.     Discussion, paragraph 3, line 4: “(Figure 3C)” should be “(Figure 4C)”.

Author Response

Dear editors and reviewer 1:

Thank you for your letter and for the reviewers’ comments concerning our manuscript entitled “Quantitative Evaluation of In Vivo Corneal Biomechanical Properties after SMILE and FLEx by Acoustic Radiation Force Optical Coherence Elastography” (ID: sensors-2101039). Those comments are all valuable and very helpful for revising and improving our paper, as well as the important guiding significance to our research. We have studied the comments carefully and have made a correction which we hope meet with approval. Revised portions are marked in red in the paper. The detailed responses to the reviewers’ comments are listed below.

Responses to the reviewers’ comments:

Comments and Suggestions for Authors:

Zhao and colleagues evaluated corneal biomechanical properties following femtosecond laser-assisted corneal refractive surgery in a rabbit model using acoustic radiation force optical coherence elastography. The methods are appropriate and clear and the results support the conclusions. The manuscript has been formatted and written very well. Although the outcome was expected theoretically and based on mathematical models, this study provides additional data using the ARF-OCE system that adds to the current knowledge.

Response: We greatly appreciate your pertinent suggestions concerning our manuscript. We have read the comments carefully and have discussed the insufficiency of the paper and the aspects should be further studied. Accordingly, we have made some changes to the manuscript. We hope that we have now produced a better account of our work.

General comment:

The relationship between the elastic modulus (E) and the tissue stiffness requires further explanation and clarification. For instance, Fig. 4 shows a decreasing E towards the posterior stroma. Since we know that the anterior stroma has higher mechanical strength, how would you explain this finding?

Response: The stroma, composed of fibrin, is the main component of the cornea, and fibrin is available in both stretched and coiled states. The tensile and swelling tests measured corneal tissue in the stretched state and showed that the anterior corneal stroma had a higher modulus of elasticity. For the OCE experiments on normal corneas, the fibrin remained in the stretched state under normal intraocular pressure loading, and the anterior matrix had a higher modulus of elasticity (Figure 4C), consistent with the results of the stretch test. However, for corneal tissue after refractive surgery, the anterior stroma was cut in a coiled state and could not bear the intraocular pressure, thus showing a lower elastic modulus in the OCE experiment.

Major comments:

  1. Please briefly discuss the advantages and disadvantages of the ARF-OCE compared with Brillouin microscopy and other in vivo elastography techniques in the Introduction (e.g., methods, reliability, feasibility, potential application in the human eye, …).

Response: For in vivo measurement of the corneal biomechanical properties, the elastography techniques were developed, such as Brillouin microscopy, ultrasound elastography. Brillouin microscopy was used to measure the compression modulus of the cornea with a high resolution based on the longitudinal wave speed. However, Brillouin signal acquisition requires long scanning times, which limits the clinical translation of Brillouin microscopy for the detection of corneal elasticity in the human eye. Ultrasound elastography has been used to quantify the elastic modulus of ocular tissues, such as the cornea and retina. However, the resolution of ultrasound elastography is on the sub-millimeter scale, and there are limitations in quantifying the elasticity of the stratified structure of the human cornea. ARF-OCE benefits from the advantages of optical coherence chromatography technology, which can realize high-resolution, high-sensitivity, rapid three-dimensional elastography of corneal tissue, and has already undergone initial clinical translation in human corneal elastography. The discussion has been added in lines 50-61, page 2 of the manuscript.

  1. Is there any test-retest variability data for the device used in this study?

Response: In our previous work, we evaluated the stability and reliability of the developed ARF-OCE system using a homogeneous agar model, and the experimental results were consistent with the mechanical tensile test results. Therefore, we have referred to this result in this study. The discussion has been marked in lines 139-140, page 5 of the manuscript.

  1. I suggest using a similar color coding for Figures4-6 so that the changes can be compared between the three models.

Response: The corneal elastic modulus ranges from 290-63 kPa after refractive surgery, which is much different from the preoperative cornea, so different color coding was used to better characterize this change. We will use a more reasonable color coding to map this results in our subsequent work.

Minor comments:

  1. Abstract, line 3: “… completeness of corneal biomechanical properties”; completeness should be replaced with a more appropriate word.

Response: The “intact” is used to replace the “completeness”. The discussion has been added in line 3 of abstract part, page 1 of the manuscript.

  1. Abstract, line 8: RSB should be defined on the first appearance in the abstract.

Response: The RSB is indicate the “residual stroma bed”. The discussion has been added in line 8 of abstract part, page 1 of the manuscript.

  1. Introduction, paragraph 1, line 5: “Shortened operation time compared to previous methods of refractive surgery”, this sentence is incomplete. Please revise.

Response: Both the flap and the refractive lenticule were created in a single surgery using the femtosecond laser, which shortens the procedure time compared to previous refractive surgery methods. The discussion has been added in lines 34-36, page 1 of the manuscript.

  1. Section 2.2., line 12: please add the approximate central lenticule thickness as well.

Response: The central lenticule thickness is about 100

μm. The discussion has been added in line 94, page 3 of the manuscript.

  1. Please add the relevant explanations to the legends of Figures 1 and 2.

Response: The relevant explanations to the legends of Figure1 and 2 were added. The discussion has been added in lines 102-106, page 3 of the manuscript, and in lines 115-120, page 4 of the manuscript.

  1. Figure 2. Please define PBS.

Response: The PBS is indicated the phosphate-buffered saline. The added discussion has been added in lines 140-142, page 5 of the manuscript.

  1. Section 3.3., line 5: “SLP” is not a known abbreviation for the slit-lamp and can be removed.

Response: We have modified this expression. The expression has been marked in line 108, page 4; line 220, page 7; and line 232, page 8 of the manuscript.

  1. Discussion, paragraph 3, line 4: “(Figure 3C)” should be “(Figure 4C)”.

Response: We have modified this error. The expression has been marked in line 278, page 10 of the manuscript.

Once again, thank you very much for your comments and suggestions. We hope that the revised manuscript is now suitable for publication.

The manuscript has been resubmitted to your journal. We look forward to your positive response.

Sincerely,

Yanzhi Zhao

Reviewer 2 Report

The authors measured the biomechanical properties of in vivo rabbits after SMILE and FLEx surgery using acoustic radiation force (ARF) optical coherence elastography (OCE). A ring array ultrasound transducer output ARF to induce a Lamb wave in the rabbit cornea. Then the optical coherence tomography (OCT) detected the Lamb waves. The Young’s moduli were quantified based on the Lamb wave velocities. The study would be beneficial to the ophthalmology. I have some concerns about the manuscript.

1. How did the authors immerse the eye of a living rabbit in the PBS during the OCE imaging? The details should be provided for readers’ reference.

2. A ring array ultrasound transducer was used in the measurements. Why did the authors use a ring array transducer? How many elements were there in the array? The purpose of the system design should be discussed.

3. From Table 1, the pre-surgery central corneal thickness (CCT) is close to the post-surgery value. The differences of the thickness are smaller than 10 um. However, the axial resolution of OCT is 10 um in this study. The measurements of the thickness may be not accuracy enough and, thus, the p-values may be not reliable. 

4. Why did the authors use a Lamb wave model?

5. The accuracy of the measurements of the Young’s moduli should be assessed. 

6. Are there any other researchers measure the Young’s moduli of normal rabbit corneas using OCE? What are the values of Young’s moduli in previous studies using OCE?

Author Response

Dear editors and reviewer 2:

Thank you for your letter and for the reviewers’ comments concerning our manuscript entitled “Quantitative Evaluation of In Vivo Corneal Biomechanical Properties after SMILE and FLEx by Acoustic Radiation Force Optical Coherence Elastography” (ID: sensors-2101039). Those comments are all valuable and very helpful for revising and improving our paper, as well as the important guiding significance to our researches. We have studied the comments carefully and have made a correction which we hope meet with approval. Revised portions are marked in red in the paper. The detailed responses to the reviewers’ comments are listed below.

Responses to the reviewers’ comments:

Comments and Suggestions for Authors:

The authors measured the biomechanical properties of in vivo rabbits after SMILE and FLEx surgery using acoustic radiation force (ARF) optical coherence elastography (OCE). A ring array ultrasound transducer output ARF to induce a Lamb wave in the rabbit cornea. Then the optical coherence tomography (OCT) detected the Lamb waves. The Young’s moduli were quantified based on the Lamb wave velocities. The study would be beneficial to the ophthalmology. I have some concerns about the manuscript.

  1. How did the authors immerse the eye of a living rabbit in the PBS during the OCE imaging? The details should be provided for readers’ reference.

Response: For the in vivo OCE experiment, the rabbit eyeball was squeezed within a rubber drape that serves as a container to immerse the eyeball in PBS. The added discussion has been added in lines 140-142, page 5 of the manuscript.

  1. A ring array ultrasound transducer was used in the measurements. Why did the authors use a ring array transducer? How many elements were there in the array? The purpose of the system design should be discussed.

Response: The ring array ultrasound transducer with a center frequency of 4.5 MHz with eight array elements allow for localized ARF-induced displacement or Lamb waves in the lateral direction with a penetration depth large enough to cover the entire cornea. On one hand, it is less expensive, less complex to manufacture and has a simpler hardware system than conventional ultrasound linear/phase array transducers. In addition, its dynamic focusing capability allows high-speed electrical scanning of the ultrasound beam without mechanically scanning the transducer across the cornea, achieving tunable excitation of the cornea in focus. The added discussion has been added in lines 128-131, page 4 of the manuscript.

  1. From Table 1, the pre-surgery central corneal thickness (CCT) is close to the post-surgery value. The differences of the thickness are smaller than 10 um. However, the axial resolution of OCT is 10 um in this study. The measurements of the thickness may be not accuracy enough and, thus, the p-values may be not reliable. 

Response: For the experiment, the central corneal thickness (CCT) was measured by ultrasound pachymetry for the elastic modulus calculation, and the ultrasound pachymetry is the gold standard for measuring corneal thickness in clinical practice. For each corneal sample, we perform six measurements and calculate the mean and standard deviation of CCT. The discussion has been marked in lines 170-171, page 5 of the manuscript.

  1. Why did the authors use a Lamb wave model?

Response: Lamb waves are perturbations that propagate in a medium with a thickness in the same order of magnitude as the characteristic wavelength of the mechanical wave (thin-plate type medium). These waves are typically dispersive, they are guided by the surface of the elastic plate, and they have multiple propagation modes distributed in different symmetric and antisymmetric orders. The speed of the Lamb wave is not only dependent of the body wave speeds of the elastic medium, but also in the excitation frequency, thickness of the plate, and the acoustic properties of the medium at the interphase of the top and bottom layer of the elastic plate. The cornea is the biological tissue that can be considered as a layered thin plate with mixed boundary conditions: the top surface medium is air or fluid, and the bottom surface is fluid. Therefore, the Lamb wave model allows for more accurate quantification of corneal biomechanical properties.

  1. The accuracy of the measurements of the Young’s moduli should be assessed. 

Response: In our previous work, we evaluated the stability and reliability of the developed ARF-OCE system using a homogeneous agar model, and the experimental results were consistent with the mechanical tensile test results. Therefore, we have referred to this result in this study. The discussion has been marked in lines 138-139, page 5 of the manuscript.

  1. Are there any other researchers measure the Young’s moduli of normal rabbit corneas using OCE? What are the values of Young’s moduli in previous studies using OCE?

Response: Yan Li et al. have measured the Young’s modulus of the normal rabbit cornea using the OCE system, and the value of Young’s moduli is 108 kPa, this is consist with our results. The discussion has been marked in lines 209-210, page 7 of the manuscript.

Once again, thank you very much for your comments and suggestions. We hope that the revised manuscript is now suitable for publication.

The manuscript has been resubmitted to your journal. We look forward to your positive response.

Sincerely,

Yanzhi Zhao

Reviewer 3 Report

This work is an interesting contribution to characterize the corneal biomechanical properties, addressing the main corneal surgical techniques. The manuscript is clear and easy to understand, the pictures has good quality and are fundamental to understand the work, the topics are well structured. A small correction should be done in order to improve the work, in Materials and methods section the last phrase compare 300-degree incision in FLEx with 2 mm linear incision in SMILE, for better understanding it should be converted in degree.

Author Response

Dear editors and reviewer 3:

Thank you for your letter and for the reviewers’ comments concerning our manuscript entitled “Quantitative Evaluation of In Vivo Corneal Biomechanical Properties after SMILE and FLEx by Acoustic Radiation Force Optical Coherence Elastography” (ID: sensors-2101039). Those comments are all valuable and very helpful for revising and improving our paper, as well as the important guiding significance to our researches. We have studied the comments carefully and have made a correction which we hope meet with approval. Revised portions are marked in red in the paper. The detailed responses to the reviewers’ comments are listed below.

Responses to the reviewers’ comments:

Comments and Suggestions for Authors:

This work is an interesting contribution to characterize the corneal biomechanical properties, addressing the main corneal surgical techniques.  The manuscript is clear and easy to understand, the pictures has good quality and are fundamental to understand the work, the topics are well structured. A small correction should be done in order to improve the work, in Materials and methods section the last phrase compare 300-degree incision in FLEx with 2 mm linear incision in SMILE, for better understanding it should be converted in degree.

Response: We greatly appreciate your pertinent suggestions concerning our manuscript. We have read the comments carefully and have discussed the insufficiency of the paper and the aspects should be further studied. Accordingly, we have made some changes to the manuscript. The discussion has been added in line 96, page 3 of the manuscript.

Once again, thank you very much for your comments and suggestions. We hope that the revised manuscript is now suitable for publication.

The manuscript has been resubmitted to your journal. We look forward to your positive response.

Sincerely,

Yanzhi Zhao

Round 2

Reviewer 1 Report

The authors have addressed all my comments satisfactorily and improved the manuscript. I only have a few minor comments on the revised manuscript:

1. Abstract, line 3: "the effect of the corneal cap on the intact of corneal biomechanical properties."; this sentence still does not seem to be grammatically correct. Maybe "... the corneal cap on the integrity of corneal biomechanical properties" or "... of the corneal cap on corneal biomechanical properties" would be more appropriate.

2. Abstract, line 8: "residual stroma bed" should be "residual stromal bed"

3. Introduction, paragraph 2, line 1: please redefine RSB "residual stromal bed" at the first appearance in the manuscript text.

Author Response

Dear editors and reviewer 1:

Thank you for your letter and for the reviewers’ comments concerning our manuscript entitled “Quantitative Evaluation of In Vivo Corneal Biomechanical Properties after SMILE and FLEx by Acoustic Radiation Force Optical Coherence Elastography” (ID: sensors-2101039, Round 2). Those comments are all valuable and very helpful for revising and improving our paper, as well as the important guiding significance to our researches. We have studied the comments carefully and have made a correction which we hope meet with approval. Revised portions are marked in red in the paper. The detailed responses to the reviewers’ comments are listed below.

Responses to the reviewers’ comments:

Comments and Suggestions for Authors:

The authors have addressed all my comments satisfactorily and improved the manuscript. I only have a few minor comments on the revised manuscript:
1. Abstract, line 3: "the effect of the corneal cap on the intact of corneal biomechanical properties."; this sentence still does not seem to be grammatically correct. Maybe "... the corneal cap on the integrity of corneal biomechanical properties" or "... of the corneal cap on corneal biomechanical properties" would be more appropriate.

Response: The sentence was modified to read: “To quantitatively evaluate the differences in corneal biomechanics after SMILE and FLEx surgery using an acoustic radiation force optical coherence elastography system (ARF-OCE) and to analyze the effect of the corneal cap on the integrity of corneal biomechanical properties. ” The discussion is marked in purple in line 3 of the abstract.  

  1. Abstract, line 8: "residual stroma bed" should be "residual stromal bed".

Response: The “residual stromal bed” is correct. We have modified the point, and the discussion is marked in purple in line 8 of the abstract.

  1. Introduction, paragraph 2, line 1: please redefine RSB "residual stromal bed" at the first appearance in the manuscript text.

Response: We have added this discussion in purple in line 41, page 2 of the manuscript.

Once again, thank you very much for your comments and suggestions. We hope that the revised manuscript is now suitable for publication.

The manuscript has been resubmitted to your journal. We look forward to your positive response.

Sincerely,

Yanzhi Zhao

Reviewer 2 Report

All the points from my previous review have been addressed in satisfying accuracy. I believe that the manuscript is in good shape to be published in this journal. 

Author Response

Dear editors and reviewer 3:

Thank you for your letter and for the reviewers’ comments concerning our manuscript entitled “Quantitative Evaluation of In Vivo Corneal Biomechanical Properties after SMILE and FLEx by Acoustic Radiation Force Optical Coherence Elastography” (ID: sensors-2101039, Round 2). Those comments are all valuable and very helpful for revising and improving our paper, as well as the important guiding significance to our researches. We have studied the comments carefully and have made a correction which we hope meet with approval. Revised portions are marked in purple in the paper.

Responses to the reviewers’ comments:

Comments and Suggestions for Authors:

A small correction should be done in order to improve the work, in Materials and methods section the last phrase compare 300-degree incision in FLEx with 2 mm linear incision in SMILE, for better understanding it should be converted in degree.

Response: We have modified the manuscript. The discussion has been added in line 96, page 3 of the manuscript.

Once again, thank you very much for your comments and suggestions. We hope that the revised manuscript is now suitable for publication.

The manuscript has been resubmitted to your journal. We look forward to your positive response.

Sincerely,

Yanzhi Zhao
